# Fast nonlinear integration drives accurate encoding of input information in large multiscale systems
Giorgio Nicoletti [1,2] & Daniel Maria Busiello [3,4] ✉

Biological and artificial systems encode information through complex nonlinear operations across multiple timescales. A clear understanding of the interplay between this multiscale structure and the nature of nonlinearities at play is, however, missing. Here, we study a general model where the input signal is propagated to an output unit through a processing layer via nonlinear activation functions. We focus on two widely implemented paradigms: *nonlinear summation*, where signals are first nonlinearly transformed and then combined; *nonlinear integration*, where they are combined first and then transformed. We find that fast-processing capabilities systematically enhance input-output mutual information, and nonlinear integration outperforms summation in large systems. Conversely, a nontrivial interplay between the two strategies emerges in lower dimensions as a function of interaction strength, heterogeneity, and sparsity of conections between the units. Finally, we reveal a tradeoff between input and processing sizes in strong-coupling regimes. Our results shed light on relevant features of nonlinear information processing with implications for both biological and artificial systems.

The ability to encode and process information from the external world is essential to maintain robust functioning in biological systems[1]. These goals are usually achieved through complex internal machinery that involves nonlinear operations. For example, multi-molecular reactions drive sensing and adaptation in chemical networks[2–4], gene regulatory dynamics is controlled by protein-mediated interactions leading to multistable phases corresponding to different cell fates[5,6], phase coexistence phenomena sustain noise reduction and functional organization in cellular environments[7], and complex interaction networks underlie the computational capabilities of neural populations[8,9]. As such, extracting information from a given input to generate a desired output is a fundamental problem that spans several fields, from signal processing in biochemical systems[10,11] to designing and training artificial neural networks[12]. Many of these systems share the idea that inputs need to be processed via different types of nonlinear activation functions to enable non-trivial learning tasks. Despite remarkable results, understanding the key determinants of how the type of nonlinearity shapes information processing is an active area of theoretical research[13–15]. Recent works have investigated the performance of computation instantiated by biological media, making an effort to bridge artificial and biochemical processing[16] and highlighting the pivotal role of nonlinear encoding[17–19] and multiple timescales[20–22].

Information theory provides us with tools to quantitatively study information-processing capabilities of various systems ranging from stochastic processes[23,24] to biological scenarios[25–27]. While the impact of timescales on information propagation has been understood in general frameworks[28,29], the role of internal nonlinear mechanisms remains unclear without focusing on specific models. One of the main difficulties resides in the lack of general analytical approaches, with results obtained for large systems and phenomenological models often relying on Gaussian approximations of various forms[30,31].

In terms of nonlinear processing, two paradigmatic schemes have been extensively implemented in a variety of different contexts[32–36]: *nonlinear summation*, in which incoming signals are first nonlinearly transformed and then summed before affecting the target; and nonlinear integration, in which signals are integrated before the nonlinear processing step. Although they underscore different underlying physical processes, summation and integration have often been used interchangeably to describe neural systems[37–46], especially when there is no clear biological reason to choose one over the other. A similar dichotomy is present in gene networks, where, given the absence of microscopic models substantiating either one of these two schemes, there is no consensus on the use of summation or integration, with various works

[1]Quantitative Life Sciences Section, The Abdus Salam International Center for Theoretical Physics (ICTP), Trieste, Italy. [2]ECHO Laboratory, École Polytechnique Fédérale de Lausanne, Lausanne, Switzerland. [3]Department of Physics and Astronomy, University of Padova, Padova, Italy. [4]Max Planck Institute for the Physics of Complex Systems, Dresden, Germany. ✉e-mail: danielmaria.busiello@unipd.it

reporting opposite approaches[47–50]. In these scenarios, despite their ability to describe the systems' dynamics, summation and integration may present striking differences when considering processing performances. These differences may be especially relevant in the presence of multiple timescales[28]. Even in those cases in which the implementation of non-linear summation or integration is biologically motivated, it would be crucial to understand their effect on information processing. For instance, dendrites are believed to perform integration in neural circuits[35]; in gene networks, gene-protein interactions appear in the form of a nonlinear summation when derived from first principles[49]; many adaptive dynamical systems are usually modeled via nonlinear integration schemes due to the presence of a slow feedback dynamics[32,34]; controlled stochastic chemical networks naturally implement summation along multiple reaction channels[51,52].

In this work, we study the information-theoretic features associated with nonlinear summation and integration in a generic multiscale information-processing system. To this end, we consider a possibly high-dimensional signal from an input unit, then processed by a processing unit, and finally encoded into an output unit. Interactions among the units form a general multilayer network structure that supports the propagation of the input information[28,53,54]. Crucially, each unit may operate on a different timescale and is composed of an arbitrary number of individual degrees of freedom (dofs), such as neurons in neural networks, chemical species in a signaling architecture, or genes in a genetic network. Operations between the units are implemented by an activation function that can perform nonlinear summation or integration of the incoming signals from the connected unit. We first find the exact expression for the probability distribution (pdf) of the system in different timescale regimes. Then, we employ this result to characterize the mutual information between input and output. We find that, in the absence of a processing unit, there exists a crossover from a region in which nonlinear summation is favored to one in which integration leads to higher mutual information. Crucially, we also show that the presence of an intermediate processing unit enhances encoding performances when acting on a faster timescale than the output unit. Further, we study the effect of the system's dimensionality, finding that fast nonlinear integration schemes are associated with higher input-output mutual information in large multiscale systems, emerging as the backbone of accurate processing in this scenario. On the other hand, nonlinear summation is beneficial in small systems with highly heterogeneous couplings. Finally, we show that nonlinear integration may lead to the spontaneous emergence of bimodality in the output layer even for Gaussian inputs, underpinning its role in implementing dynamical input discrimination that can be tuned by tinkering with internal parameters.

## Results
### Multiscale information-processing systems
We consider a general information-processing system with three different stochastic units: input $I$, processing $P$, and output $O$. Each unit is composed of $M_\mu$ degrees of freedom (dofs) with a shared timescale $\tau_\mu$ and whose activity is denoted by $\mathbf{x}_\mu \in \mathbb{R}^{M_\mu}$, with $\mu = I, P, O$. All dofs within the same unit are linearly coupled with an interaction matrix $\hat{A}_\mu \in \mathbb{R}^{M_\mu \times M_\mu}$; conversely, the coupling from unit $\nu$ to $\mu$ is implemented via a nonlinear activation function $\boldsymbol{\phi}_{\mu\nu} \in \mathbb{R}^{M_\mu}$ that depends, in principle, on all dofs within $\nu$ and an interaction matrix $\hat{A}_{\mu\nu} \in \mathbb{R}^{M_\mu \times M_\nu}$. The system's dynamics is described by the following set of Langevin equations:

$$\tau_\mu \dot{\mathbf{x}}_\mu = -\hat{A}_\mu \mathbf{x}_\mu + \sum_{\nu \neq \mu} g_{\mu\nu} \boldsymbol{\phi}_{\mu\nu}(\hat{A}_{\mu\nu}; \mathbf{x}_\nu) + \sqrt{2\tau_\mu} \hat{\sigma}_\mu \boldsymbol{\xi}_\mu, \quad (1)$$

where $g_{\mu\nu}$ is the interaction strength between unit $\nu$ and $\mu$, $\boldsymbol{\xi}_\mu$ a vector of Gaussian white noises, and $\hat{D}_\mu = \hat{\sigma}_\mu \hat{\sigma}_\mu^T \in \mathbb{R}^{M_\mu \times M_\mu}$ defines a diagonal diffusion matrix. For simplicity, we take $\hat{D}_\mu$ to be the identity matrix. We also assume that the input evolves independently, as this is the case for a relevant class of biophysical scenarios[31,55–57]. Then, the input is passed to the processing unit through a directional coupling, i.e., $g_{PI} \neq 0$ and $g_{IP} = 0$. After the

processing step, the signal arrives at the output unit, again through a directional coupling, i.e., $g_{OP} \neq 0$ and $g_{PO} = 0$.

To investigate how the mechanisms implementing internal nonlinear processing affect the information content of the system, we study the mutual information between input and output units,

$$
\begin{aligned}
I_{IO} &= \int d\mathbf{x}_I d\mathbf{x}_O p_{IO}(\mathbf{x}_I, \mathbf{x}_O) \log_2 \frac{p_{IO}(\mathbf{x}_I, \mathbf{x}_O)}{p_I(\mathbf{x}_I) p_O(\mathbf{x}_O)} \\
&= H_O - \int d\mathbf{x}_I p_I(\mathbf{x}_I) h_{O|I}(\mathbf{x}_I) = H_O - \langle h_{O|I} \rangle_I,
\end{aligned} \quad (2)
$$

where $h_{O|I} = \langle p_{O|I} \log_2 p_{O|I} \rangle_O$. Here, $p_{IO}(\mathbf{x}_I, \mathbf{x}_O)$ is the joint pdf of input and output dofs, $p_I(\mathbf{x}_I)$ and $p_O(\mathbf{x}_O)$ are their respective marginal pdfs, and $H_O$ is the Shannon entropy[58] of the output dofs distribution computed in bits. $I_{IO}$ quantifies the information shared between $I$ and $O$, therefore acting as an unbiased proxy for processing accuracy in this paradigmatic setting[58].

As demonstrated in[28], if the dynamics of the input unit are the fastest at play ($\tau_I \ll \tau_P, \tau_O$), no mutual information can be generated between $I$ and $O$. Conversely, a slow input is a necessary condition to have a non-zero $I_{IO}$. We still have the freedom to set the timescales of processing and output units, distinguishing two relevant cases: a fast-processing system ($\tau_P \ll \tau_O$) and a slow-processing one ($\tau_P \gg \tau_O$). However, a crucial role is also played by the specific type of nonlinearity at hand, encoded in the vector $\boldsymbol{\phi}_{\mu\nu}$. As discussed in the introduction, we distinguish two widely used but distinct cases, corresponding to different processing schemes: nonlinear summation (ns)[36,39,40,45,46,59] and integration (int)[32,34,35,37,38,60–62]. By setting a hyperbolic tangent as activation function, a customary modeling choice for recurrent neural networks[36,45,59], we have the following forms for the $i$-th component of interaction terms between units:

$$
\begin{aligned}
(\phi_{\mu\nu}^i)^{\text{ns}} &= \frac{1}{M_\nu} \sum_{j=1}^{M_\nu} A_{\mu\nu}^{ij} \tanh(x_\nu^j) \\
(\phi_{\mu\nu}^i)^{\text{int}} &= \tanh\left(\frac{1}{M_\nu} \sum_{j=1}^{M_\nu} A_{\mu\nu}^{ij} x_\nu^j\right),
\end{aligned} \quad (3)
$$

where all nodes in unit $\nu$ contribute to the dynamics of node $i$ in unit $\mu$ through the nonlinear activation function and the set of weights $A_{\mu\nu}^{ij}$, with $j = 1, \ldots, M_\nu$, mediating the coupling. These two cases represent different physical processes. For a nonlinear summation, the signals generated by each dof in unit $\nu$ are first nonlinearly transformed, and then linearly projected by means of the interaction matrix $\hat{A}_{\mu\nu}$. In contrast, for nonlinear integration, the signals from unit $\nu$ are first linearly combined via the weight matrix $\hat{A}_{\mu\nu}$, and then the resulting integrated signal is nonlinearly transformed by the activation function and passed to the $i$-th dof of unit $\mu$.

### Exact solution for fast and slow processing units
The first contribution of this work is to provide an analytical solution for the joint distribution of the whole system, $p_{IPO}$, that can be exploited to evaluate the input-output mutual information $I_{IO}$, and the output pdf $p_O$. While $I_{IO}$ informs us on the processing performance of the system, $p_O$ contains information on the ability to perform input discrimination. $p_{IPO}$ satisfies the following Fokker-Planck equation[63]:

$$\frac{\partial}{\partial t} p_{IPO} = \left(\frac{\mathcal{L}_I}{\tau_I} + \frac{\mathcal{L}_P}{\tau_P} + \frac{\mathcal{L}_O}{\tau_O}\right) p_{IPO} \quad (4)$$

where $\mathcal{L}_\mu$ is the Fokker-Planck operator associated with the unit $\mu = I, P, O$, as detailed in the Supplementary Notes 1 and 2. Although general exact expressions are out of reach without approximations, the limits of fast and slow processing can provide useful insights into system operations, provided the presence of a slow input unit. From[28,29], we know that in these two limiting regimes the joint pdf of input, processing, and output dofs is the product of conditional distributions. As we show in the "Methods" and the Supplementary Note 3 and 4, at the steady-state (i.e., when $\partial_t p_{IPO} = 0$) we

have the steady-state or stationary distributions:

$$p_{IPO}^{\text{fp}} = p_I^{\text{st}} p_{P|I}^{\text{st}} p_{O|I}^{\text{eff,st}} \qquad \text{fast processing} \qquad (5)$$

$$p_{IPO}^{\text{sp}} = p_I^{\text{st}} p_{P|I}^{\text{st}} p_{O|P}^{\text{st}} \qquad \text{slow processing ,} \qquad (6)$$

where the superscript "st" (omitted on the l.h.s.) stands for stationary, and "eff" indicates a pdf that solves an effective operator obtained from the ensemble average over dofs faster than its corresponding unit. We use the superscript "fp" and "sp" to indicate that these quantities are evaluated respectively for fast and slow processing. Let us inspect all these terms one by one. $p_I^{\text{st}}$ is the multivariate Gaussian distribution of the input dofs with mean $\mathbf{m}_I$ and covariance matrix $\hat{\Sigma}_I$ that solves the Lyapunov equation $\hat{A}_I \hat{\Sigma}_I + \hat{\Sigma}_I \hat{A}_I^T = 2\hat{D}_I$. By exploiting the fact that intra-unit interactions are linear, all the conditional distributions may be written as:

$$p_{\mu|\nu}^{\text{st}} = \mathcal{N}_\mu \left( \mathbf{m}_{\mu|\nu}(\mathbf{x}_\nu), \hat{\Sigma}_\mu \right) \qquad \mu, \nu = I, P, O \qquad (7)$$

with $\mathcal{N}_\mu$ a Gaussian distribution over $\mathbf{x}_\mu$, $\hat{\Sigma}_\nu$ satisfying its corresponding Lyapunov equation, and the average containing the dependence on the conditional variable as follows:

$$\mathbf{m}_{\mu|\nu}(\mathbf{x}_\nu) = g_{\mu\nu} \hat{A}_\mu^{-1} \boldsymbol{\phi}_{\mu\nu}(\hat{A}_{\mu\nu}; \mathbf{x}_\nu) . \qquad (8)$$

Notice that the functional form of Eq. (8) depends on the nonlinear processing mechanism considered in Eq. (3). However, when an effective operator is involved, calculations become harder. By using a convergent expansion of the hyperbolic tangent, we show that:

$$p_{O|I}^{\text{eff,st}} = \mathcal{N}_O \left( \mathbf{m}_{O|I}^{\text{eff}}(\mathbf{x}_I), \hat{\Sigma}_O \right) \qquad (9)$$

with again $\hat{A}_O \hat{\Sigma}_O + \hat{\Sigma}_O \hat{A}_O^T = 2\hat{D}_O$ and

$$\text{ns}: \quad \mathbf{m}_{O|I}^{\text{eff}} = g_{OP} \hat{A}_O^{-1} \left( \frac{\hat{A}_{OP}}{M_P} \mathcal{F}(\mathbf{m}_{P|I}, \text{diag}(\hat{\Sigma}_P)) \right) \qquad (10)$$

$$\text{int}: \quad \mathbf{m}_{O|I}^{\text{eff}} = g_{OP} \hat{A}_O^{-1} \mathcal{F}(\mathbf{m}_{\text{int}}, \mathbf{v}_{\text{int}})$$

where we employed the shorthand notation $\mathcal{F}^i(\mathbf{x}, \mathbf{y}) = \mathcal{F}(x^i, y^i)$. In particular, $\mathcal{F}$ is a nontrivial nonlinear function defined in the "Methods" and in the Supplementary Note 4, and we introduced the following integrated quantities:

$$\mathbf{m}_{\text{int}} = \frac{1}{M_P} \hat{A}_{OP} \mathbf{m}_{P|I} , \qquad \mathbf{v}_{\text{int}} = \frac{1}{M_P^2} \hat{A}_{OP} \hat{\Sigma}_P \hat{A}_{OP}^T . \qquad (11)$$

From Eq. (10), we notice that the dependence on $\mathbf{x}_I$ enters solely through $\mathbf{m}_{O|P}$, defined in Eq. (8). The main difference resides in the fact that, in the case of summation, the nonlinear function $\mathcal{F}$ has to be averaged with processing weights $\hat{A}_{OP}$, while in the case of integration, $\mathcal{F}$ must be directly evaluated on integrated quantities.

Putting all these results together, we obtain an analytical expression for the joint pdf of the whole system, $p_{IPO}$. We stress that $p_{IPO}$ is a highly nonlinear distribution. However, our factorization into conditional Gaussian distributions incorporates the nonlinearities only into their means, allowing in particular for efficient sampling (see "Methods"). Furthermore, the structure of the resulting conditional dependencies is crucially different between fast and slow processing units, with fundamental implications for the mutual information between the input and the output. To obtain general results, we focus on the case of random interactions, an approach that has provided fundamental insights in several fields[32,33,36,59,64–66]. We take interactions within the same unit $\mu$ to be distributed as $A_\mu^{ij} \sim \mathcal{N}(0, \sigma_\mu/\sqrt{M_\mu})$ with diagonal elements $A_\mu^{ii} = 1$ for all $i = 1, \dots, M_\mu$, so that, as unit

dimensions increase, it remains linearly stable if $\sigma_\mu < 1$ (see ref. 67 and Supplementary Note 3). Interactions from unit $\nu$ to $\mu$ are instead distributed as $A_{\mu\nu}^{ij} \sim \mathcal{N}(0, \sigma_{\mu\nu})$, and all results are obtained by averaging over realizations of these random matrices. Intuitively, while $g_{\mu\nu}$ describes the overall interaction strength from $\nu$ to $\mu$, $\sigma_{\mu\nu}$ models the intrinsic coupling heterogeneity.

In Fig. 1, we show stochastic trajectories and probability distributions at the steady state of the output degree of freedom for slow and fast processing, both in the case of summation and integration. For simplicity of computation and visualization, we will consider a one-dimensional output unit throughout this manuscript. While there is no striking difference between slow and fast processing at the dynamical level, nonlinear summation and integration lead to two very different distributions in the output node. Integrating incoming signals from one unit to the other favors the spontaneous emergence of a pronounced switching behavior that reflects into a bimodal distribution, a signature of input discrimination. The last part of this manuscript will be dedicated to quantitatively substantiating this observation.

## Enhanced information by fast processing units

We can now exploit the exact factorization of the joint pdf of the system to evaluate the accuracy of processing the stochastic input and encoding it into the one-dimensional output, by means of the mutual information $I_{IO}$ in Eq. (2). To establish a baseline for the full processing scheme described in the previous section, we first consider the simpler scenario of an input signal $\mathbf{x}_I$ that is directly passed to a one-dimensional output unit $\mathbf{x}_O$. Once again, we focus on the limiting case of a slow input ($\tau_I \gg \tau_O$) in which information can be transferred from the input to the output unit[28,29]. The joint steady-state distribution of input and output dofs reads $p_{IO}^{\text{np}} = p_I^{\text{st}} p_{O|I}^{\text{np,st}}$, where the superscript "np" stands for "no processing". Here, $p_{O|I}^{\text{np,st}}$ is a Gaussian distribution whose variance is independent on $\mathbf{x}_I$ (see Eq. (7) and "Methods" for details). Thus, $h_{O|I}^{\text{np}}$ does not depend on $\mathbf{x}_I$ and is equal to

$$h_{O|I}^{\text{np}} = \frac{1}{2} \left[ M_O(1 + \log_2(2\pi)) + \log_2 \det(\hat{\Sigma}_O) \right] \qquad (12)$$

so that the mutual information simply reads

$$I_{IO}^{\text{np}} = H_O^{\text{np}} - h_{O|I}^{\text{np}} . \qquad (13)$$

Therefore, evaluating $I_{IO}^{\text{np}}$ amounts to computing the Shannon entropy of the output distribution, which can be done using standard estimators (see "Methods")[68].

In Fig. 2a–c, we compare the behavior of $I_{IO}^{\text{np}}$ with nonlinear summation and nonlinear integration, as a function of the input-output coupling strength, $g_{OI}$, and the standard deviation of their interactions, $\sigma_{OI}$, that accounts for coupling heterogeneity. As expected, in both cases information increases with $g_{OI}$. Crucially, we also find that, while nonlinear integration performs better at small $\sigma_{OI}$, nonlinear summation becomes dominant at large $\sigma_{OI}$. This nontrivial switch signals the fact that, in the presence of large elements in the interaction matrix, $I_{IO}^{\text{np}}$ is favored by nonlinear summation. Additionally, as shown in Fig. 2d, e, the output distribution with nonlinear integration becomes bimodal for large $\sigma_{OI}$ due to the saturating effect of the hyperbolic tangent—a phenomenon much more pronounced when all the inputs are summed together in the argument of the activation function. We also find (Fig. 2f, g) that mutual information saturates to a finite value as the input gets closer to linear instability, a feature already observed in models of neural populations[19]. In particular, an input closer to linear instability appears to be always beneficial for nonlinear summation, further suggesting that large input values—either from $\mathbf{x}_I$ itself or due to specific large couplings—are better represented in the output by summing separate activation functions. With nonlinear integration, instead, linear instability and large values of the input may decrease $I_{IO}^{\text{np}}$, as they may push the activation function to saturation.

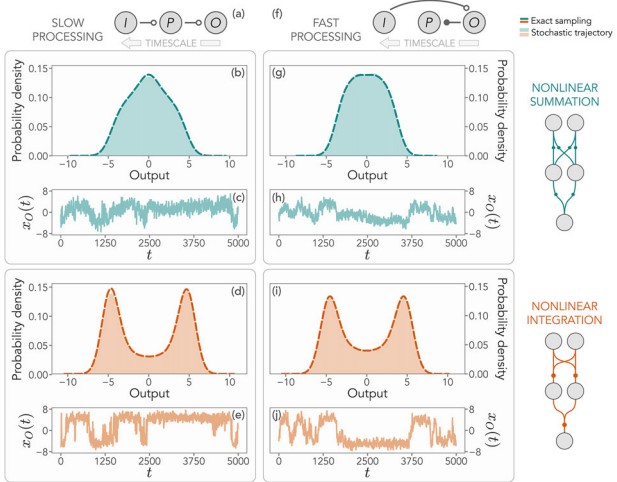

**Fig. 1 | Output distribution and trajectories for different nonlinear schemes and timescale orderings. a** Scheme of the model for a slow processing unit. From left to right, units are ordered with decreasing timescales. $I$ indicates the input, $P$ the processing unit, and $O$ the output. Links with empty dots denote interactions from a slow to a fast unit. **b** Output distribution for a slow processing unit in the presence of nonlinear summation, where the activity of the components of each unit is first integrated and then summed. The dashed line is obtained by exact sampling, while the shaded area represents the histograms obtained from Langevin trajectories with a timescale ratio $\Delta\tau = 10^{-2}$ between the units. **c** The stochastic trajectory of the one-dimensional output unit in this regime. **d** Same as (**b**) in the presence of nonlinear integration. **e** Stochastic trajectory of the one-dimensional output in this case. **f** Scheme of the model for a fast processing unit, with the same ordering as in (**a**). The link with a filled dot denotes interactions from a faster to a slower unit. **g** Output distribution for a fast processing unit in the presence of nonlinear summation, where the activity of the components of each unit is first summed and then integrated. **h** Stochastic trajectory for the one-dimensional output in this case. **i** Same as panel (**g**) in the presence of nonlinear integration. **j** Stochastic trajectory for the one-dimensional output in this case. In this figure, the unit dimensions are $M_I = 5$, $M_P = 3$, and $M_O = 1$. Interactions between units are distributed as $\mathcal{N}(0, 1)$, whereas intra-unit interactions follow $\mathcal{N}(0, 0.9/\sqrt{M_\mu})$.

Then, we add back a processing unit to understand its effect on the mutual information between input and output units. We note that the system with or without processing is fundamentally different, as the processing unit evolves on its own timescale and therefore alters both the form of the Fokker-Planck operators and the structure of the associated joint probability distributions. In the case of fast processing, the joint steady-state distribution of input and output dofs is obtained from Eq. (5) by integrating over $\mathbf{x}_P$, i.e., $p_{IO}^{\mathrm{fp}} = p_I^{\mathrm{st}} p_{O|I}^{\mathrm{eff,st}}$. Thus, as before, the variance of the Gaussian distribution $p_{O|I}^{\mathrm{eff,st}}$ is independent of $\mathbf{x}_I$ (Eq. (9)), and the mutual information can be written following Eqs. (12) and (13) (using fp as a superscript). In the presence of a slow processing unit, instead, from Eq. (6) we have:

$$p_{IO}^{\mathrm{sp}} = p_I^{\mathrm{st}} \int d\mathbf{x}_P \, p_{O|P}^{\mathrm{st}} p_{P|I}^{\mathrm{sp,st}} := p_I^{\mathrm{st}} p_{O|I}^{\mathrm{sp,st}} . \tag{14}$$

Although an expression for $h_{O|I}^{\mathrm{sp}}$ cannot be easily obtained in this case, we can efficiently sample $p_{O|I}^{\mathrm{sp,st}}$ by using Eq. (14) to compute the mutual information $I_{IO}^{\mathrm{sp}}$ from Eq. (2), as detailed in the "Methods". In order to compare the results with the ones in the absence of a processing unit, we replace the previous output with a one-dimensional processing, i.e., we take $g_{OI} \to g_{PI}$ and $\sigma_{OI} \to \sigma_{PI}$. Then, we add an additional one-dimensional layer that now constitutes the output of the system. In this way, by removing the processing unit, we get back the original input-output system, allowing a direct comparison between the two. We study the behavior of the mutual

information as we change the processing-output coupling, $g_{OP}$, and interaction heterogeneity, $\sigma_{OP}$.

In Fig. 2h–j we show that, with a fast processing unit, for sufficiently large $g_{OP}$ the mutual information $I_{IO}^{\mathrm{fp}}$ is larger than that of the corresponding input-output system. The coupling value for which such crossing takes place decreases with $\sigma_{OP}$, hinting at the fact that a system with either large $g_{OP}$ or $\sigma_{OP}$ can outperform its counterpart without processing. On the other hand, this is not possible with a slow processing unit, for which the presence of a one-dimensional processing layer seems to be detrimental, or immaterial at best (Fig. 2k–m). Indeed, we find that $I_{IO}^{\mathrm{sp}} \leq I_{IO}^{\mathrm{np}}$, approaching this value with nonlinear summation at large $g_{OP}$. Once more, whether nonlinear integration or summation leads to more accurate input encoding primarily depends on $g_{OP}$ and $\sigma_{OP}$.

## Enhanced information by nonlinear integration

While a one-dimensional processing can be advantageous or detrimental depending on its timescale, increasing its dimensionality can modify this picture. We now explore this direction, starting with the case in which processing and input units have the same large dimension $M_I = M_P = 50$. In Fig. 3, we compare the mutual information between input and output for the case of nonlinear summation, $I_{IO}^{\mathrm{ns}}$, and integration, $I_{IO}^{\mathrm{int}}$, for both slow and fast processing. We first take $\sigma_{PI} = \sigma_{OP} = 1$ to study the effects of the couplings $g_{PI}$ and $g_{OP}$. Figure 3a–c and d–f, respectively show that, independently of the internal timescale ordering, nonlinear integration leads to higher mutual information with respect to summation. Notice that, as in the previous one-dimensional case, for both nonlinear schemes we find that a fast processing unit systematically outperforms a slow one (see Fig. 3g–h). Therefore, from now on, we will only consider the fast-processing scenario. Furthermore, in Fig. 3i–j, we show that $I_{IO}^{\mathrm{int}}$ displays a nontrivial peak as a function of $g_{PI}$, whereas $I_{IO}^{\mathrm{ns}}$ saturates to lower values. This observation signals the existence of an optimal value of input-output coupling that helps maximize the encoding performance with (fast) high-dimensional processing units and nonlinear integration.

So far, we have focused on the case of small heterogeneity of processing-output interactions, where nonlinear integration displays a computational advantage. However, if we keep both $M_P$ and $M_I$ fixed, the situation is reversed at larger $\sigma_{OP}$ and large $g_{PI}$ (Fig. 3k, l). In this regime, $I_{IO}^{\mathrm{ns}}$ saturates at values that are larger than the peak of $I_{IO}^{\mathrm{int}}$. This suggests once more the presence of a nontrivial interplay between the two schemes as a function of interaction heterogeneity and coupling strengths.

Crucially, even in these strong-coupling and large-heterogeneity regimes, the computational advantage of nonlinear integration is restored at sufficiently large processing sizes (Fig. 4a–d). Intuitively, for small $M_P$, this may be due to large elements of the interaction matrices affecting only a few terms in the nonlinear summation scheme. On the other hand, the same elements may push nonlinear integration into the saturation regime if not balanced by any opposite signals, therefore masking the other interactions. Overall, we find that nonlinear summation provides higher mutual information at small $M_P$, and with large couplings and heterogeneity, but this effect becomes less and less prominent and eventually disappears as $M_P$ increases (Fig. 4e). Importantly, this effect depends on the sparsity of the connections between the units. In this case, we rewrite Eq. (3) as

$$(\phi_{\mu\nu}^i)^{\mathrm{ns}} = \frac{1}{C_{\mu\nu}^i} \sum_{j=1}^{M_\nu} A_{\mu\nu}^{ij} \tanh(x_\nu^j)$$
$$(\phi_{\mu\nu}^i)^{\mathrm{int}} = \tanh\left(\frac{1}{C_{\mu\nu}^i} \sum_{j=1}^{M_\nu} A_{\mu\nu}^{ij} x_\nu^j\right), \tag{15}$$

where $C_{\mu\nu}^i$ is the number of connections from unit $\nu$ to the $i$-th node of unit $\mu$. In Fig. 4e, we show that increasing this sparsity by reducing the probability of connection between the units, $p_{\mathrm{unit}}$, is qualitatively equivalent to effectively reducing the processing unit dimension, as expected. Hence, for

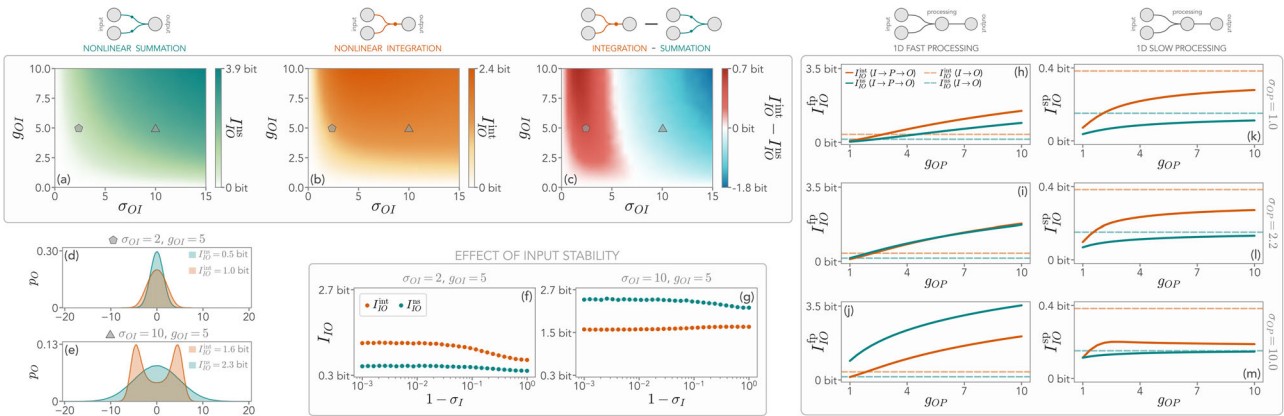

**Fig. 2 | Information-theoretic differences between nonlinear summation and integration with and without a one-dimensional processing unit. a–c** Mutual information between input and output $I_{IO}$ in a system without a processing unit, as a function of the coupling strength $g_{OI}$ and the standard deviation $\sigma_{OI}$ of the interaction matrix $\hat{A}_{OI}$ for nonlinear summation (superscript "ns", teal), nonlinear integration (superscript "int", orange) and their difference in (**c**). At small $\sigma_{OI}$, information is higher in the presence of nonlinear integration, whereas at large $\sigma_{OI}$, nonlinear summation wins. The pentagon and triangle represent specific parameter values analyzed in (**d–e**). The sketch on top of the panels represent different computational schemes. (**d-e**) Probability distribution of the one-dimensional output, $p_O$, for the specific values highlighted in (**a–c**). At large $\sigma_{OI}$, the output distribution may become bimodal with nonlinear integration, due to the saturation of the

activation function for large arguments. **f, g** Mutual information $I_{IO}$ as a function of the input stability for two different choices of parameters. As the input gets closer to linear instability, $\sigma_I \to 1$, the mutual information grows for nonlinear summation. For nonlinear integration, instead, it saturates at possibly lower values at large $\sigma_{OI}$. **h–j** Mutual information $I_{IO}$ with the addition of a one-dimensional processing unit evolving on a fast timescale for different values of interaction heterogeneity $\sigma_{OP}$. For sufficiently large processing-output couplings $g_{OP}$, the information (solid lines) is larger than the corresponding input-output system, where the processing is removed (dashed lines). Here, $g_{PI} = 2$. (**k-m**) Same, with a processing unit slower than the output. Now, the mutual information is always smaller than in the input-output system. In this Figure, unless otherwise specified, the input dimension is $M_I = 50$ and its interaction heterogeneity is $\sigma_I = 0.9$.

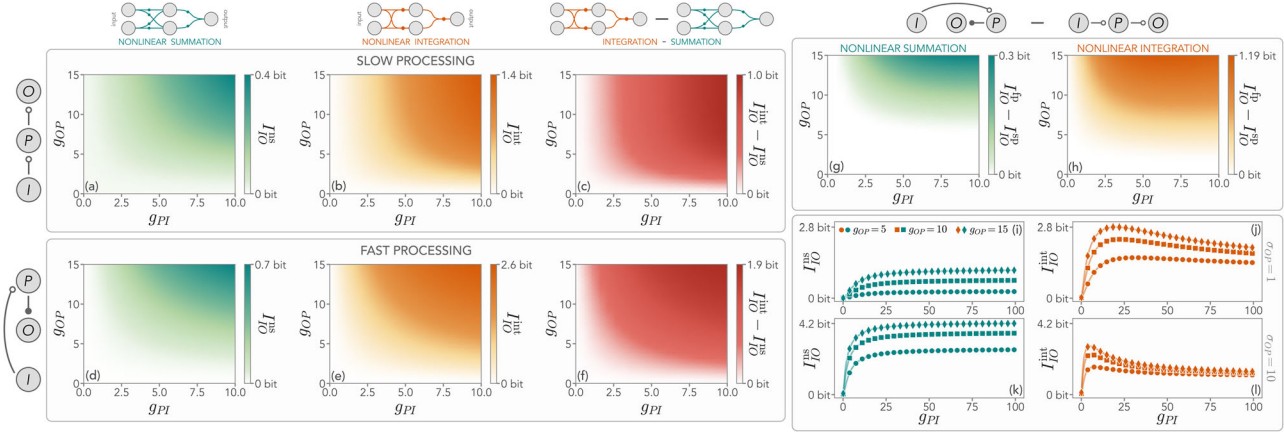

**Fig. 3 | Information-theoretic differences between nonlinear summation and integration for fast and slow processing units. a–c** Mutual information between input and output $I_{IO}$ in a system with a slow processing unit as a function of the coupling strengths $g_{PI}$ and $g_{OP}$ for nonlinear integration (superscript "int", orange), nonlinear summation (superscript "ns", teal), and their difference in (**c**). Nonlinear integration produces higher information than nonlinear summation. The sketch on the left of (**a**) indicates a system with a slow processing unit, as explained in Fig. 1a. The sketch on top of the panels represent different computational schemes.
**d–f** Same, but for a fast processing unit, with the sketch on the left of (**d**) indicating a system with a fast processing unit (see Fig. 1b). **g–h** Difference of mutual information between the fast and slow processing scenarios for nonlinear summation and

integration. $I_{IO}$ is systematically higher with a fast processing unit. This effect is particularly relevant for an activation function implementing nonlinear integration. The sketch on top graphically represents the difference between the two schemes shown in here. **i, j** Mutual information for nonlinear summation and integration at fixed $\sigma_{OP} = 1$. Integration always outperforms nonlinear summation, and displays a nontrivial peak of $I_{IO}$ at intermediate values of $g_{PI}$. **k, l** Same as (**i–j**) but for $\sigma_{OP} = 10$. The situation is reversed in this case, with nonlinear summation leading to a larger mutual information. In this figure, unless otherwise specified, the standard deviations of the interaction matrices are $\sigma_{PI} = \sigma_{OP} = 1$, $\sigma_I = \sigma_P = 0.9$, and the input and processing dimensions are $M_I = M_P = 50$. Results are obtained by averaging over $10^3$ realizations of the random interaction matrices.

fixed $M_P$, a system with sparser inter-unit connections will favor nonlinear summation, in line with our previous considerations. Furthermore, our results are qualitatively robust with respect to changes in the processing unit topology. In particular, in Fig. 4f, we consider a Barabasi-Albert topology with Gaussian weights for $\hat{A}_P$, to introduce a processing structure with hubs and a nontrivial degree distribution. Although this sparser intra-unit topology tends to favor nonlinear integration, we find in general the same interplay between the two nonlinear schemes as in the fully

connected case. In the Supplementary Note 4, we show that these results remain robust in the presence of other topologies of the processing unit, as well as in the case where $p_{unit}$ is not constant but varies across nodes. While our results remain limited to a set of paradigmatic topologies, the observed robustness seems to indicate the fundamental role of key topological parameters, such as the degree of the processing layer and the number of inter-layer connections, in determining the processing performance of complex nonlinear systems.

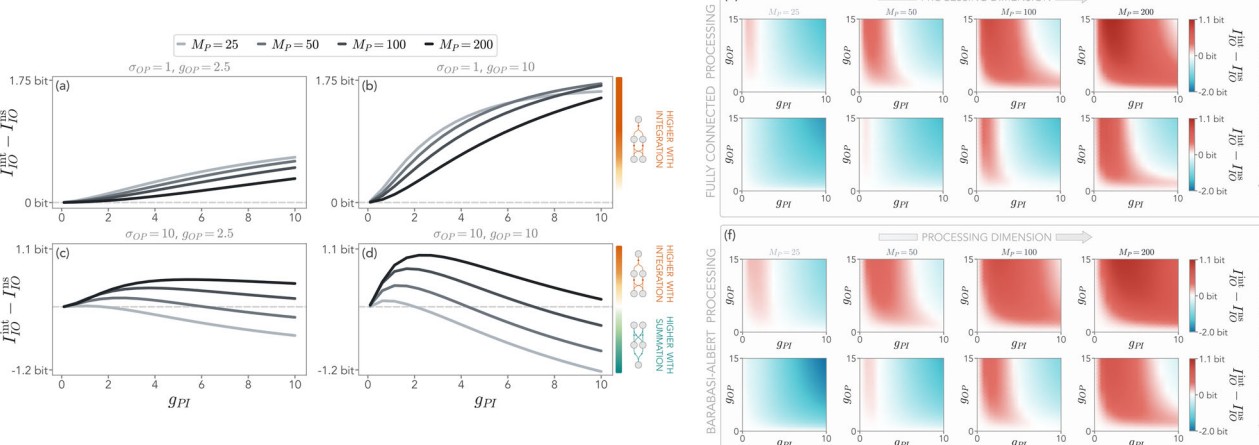

**Fig. 4 | Dependence of the input–output mutual information on the processing dimension and topology. a**, **b** Difference between the mutual information between the input and the output $I_{IO}$ for nonlinear integration and nonlinear summation as a function of the coupling between the input and the processing $g_{PI}$ for two different choices of parameters. In a fast-processing system, the advantage of nonlinear integration depends on the interplay between the dimensionality of the processing unit, $M_P$, and the heterogeneity of $\hat{A}_{OP}$, quantified by $\sigma_{OP}$. For small values ($\sigma_{OP} = 1$), we always find $I_{IO}^{int} > I_{IO}^{ns}$. **c**, **d** At large heterogeneity ($\sigma_{OP} = 10$), instead, such an advantage is achieved only for large enough $M_P$, and depends on both couplings $g_{PI}$ and $g_{OP}$. The sketches next to **b** and **d** represent nonlinear integration (orange) and nonlinear summation (teal), and the intensity of the colorbar quantifies the advantage of one over the other. **e** Difference of mutual information between nonlinear integration and summation as a function of coupling strengths $g_{PI}$ and $g_{OP}$, for different processing dimensions and sparsity of inter-unit couplings. We are

considering a fully connected processing unit. In general, increasing the processing dimension $M_P$ determines whether nonlinear integration or nonlinear summation leads to a higher mutual information between the input and the output. If $M_P$ is small, $I_{IO}^{ns}$ is typically higher in a strong-coupling regime, whereas $I_{IO}^{int} > I_{IO}^{ns}$ at large $M_P$. Increasing the sparsity of the connections between units effectively reduces the processing dimension (here, we set the probability of a connection between nodes of different units to $p_{unit} = 0.5$). **f** Same as (**e**), but changing the topology of the processing unit. We observe that this modification does not qualitatively affect the results, although more sparse processing networks favor nonlinear integration. In this Figure, the standard deviations of the interaction matrices are $\sigma_{PI} = 1$, $\sigma_I = \sigma_P = 0.9$, the input dimension is $M_I = 50$, and the interaction heterogeneity is $\sigma_{OP} = 10$ in (**e**, **f**). Results are obtained by averaging over $10^3$ realizations of the random interaction matrices.

## Interplay between input and processing dimensionality

The results in Fig. 4 suggest that the size of the processing unit deeply affects the information between input and output dofs. To further explore this effect, we now focus on fast nonlinear integration and study the interplay between input and processing dimensionalities. In Fig. 5a, we show that, at a given $M_I$, there exists an optimal value of $M_P = M_P^*$ that maximizes $I_{IO}^{int}$. $M_P^*$ decreases by increasing the input size, so that smaller inputs are optimally processed by large processing units. This effect emerges at sufficiently strong coupling regimes, as we show in Fig. 5b and in the Supplementary Note 4. We also find that the optimal processing dimension $M_P^*$ increases with $\sigma_{OP}$ (Fig. 5c).

In particular, for small $g_{OP}$ and $\sigma_{OP}$, information is typically higher for smaller input sizes (Fig. 5d), and the optimal processing dimension $M_P^*$ remains small at any $M_I$. Slightly increasing the interaction heterogeneity $\sigma_{OP}$ while keeping fixed $g_{OP}$ drastically alters $I_{IO}^{int}$ (Fig. 5e), revealing the nontrivial interplay between $M_I$ and $M_P^*$. Heuristically, this provides evidence that a nonlinear embedding of a low-dimensional input in a higher-dimensional processing space favors information encoding. On the contrary, $I_{IO}^{int}$ is maximal at small $M_P$ for large $M_I$, so that information processing of high-dimensional inputs is favored by a nonlinear compression of the input in a lower-dimensional processing space. This behavior may reveal quantitative insights into optimal operation regimes and diverse strategies to encode information.

## Emergent output bimodality

Nonlinear integration may also be advantageous from a dynamical perspective. In Fig. 6a, we consider the case of a fast processing unit and compare the bimodality of the output pdf obtained for nonlinear summation and integration by means of the Staple's bimodality coefficient (see "Methods"), respectively denoted by $b_O^{ns}$ and $b_O^{int}$. We show that integration is associated with higher bimodality coefficients, i.e., more pronounced bimodality, thus enabling more accurate input discrimination in the output distribution. We note, however, that this effect is purely dynamic, as higher bimodality does not always imply larger input-output mutual information (see for instance Fig. 2d, e and Supplementary Note 5). The presence of a

stochastic Gaussian input and random interaction matrices makes it particularly hard to pinpoint the input features that the system is discriminating. Crucially, however, this emergent bimodality may be tuned by introducing suitable processing biases in how the signal of certain nodes is encoded. We can add this ingredient in Eq. (3) by applying the substitution $x_v^j \rightarrow x_v^j - \theta_v^j$, where the bias is introduced as:

$$
\begin{aligned}
(\phi_{\mu v}^i)^{ns,b} &= \frac{1}{M_v} \sum_{j=1}^{M_v} A_{\mu v}^{ij} \tanh\left(x_v^j - \theta_v^j\right) \\
(\phi_{\mu v}^i)^{int,b} &= \tanh\left(\frac{1}{M_v} \sum_{j=1}^{M_v} A_{\mu v}^{ij}(x_v^j - \theta_v^j)\right),
\end{aligned}
\tag{16}
$$

where the superscript "b" indicates the presence of the bias. Note that $\theta v$ can be, in principle, different, for each unit. In Fig. 6b–e, we consider a system in the presence of a fast processing unit and include the presence of a random bias $\theta_v$ whose elements are drawn from $\mathcal{N}(0, 1)$ for each $v$. By comparing the same realization of random interaction matrices $\hat{A}_\mu$ and $\hat{A}_{\mu v}$ (as discussed for Fig. 3) without (Fig. 6b, c) and with (Fig. 6d, e) $\theta_v$, we find that the presence of a bias triggers an unbalance in the output bimodality. Notice that this emergent imbalance can be different between summation and integration due to the intrinsic randomness of all system's components, as shown in Fig. 6. Thus, nonlinear integration enables more pronounced tunable output bimodality, allowing the system to statistically select one of the two emerging modes.

## Discussion

In this work, we studied nonlinear processing through different activation functions in a paradigmatic information-processing system. By leveraging the presence of multiple timescales, we analytically obtained the joint distribution of the system and computed the input-output mutual information. We compared two nonlinear processing schemes, summation and integration, which have been employed in several contexts. In systems

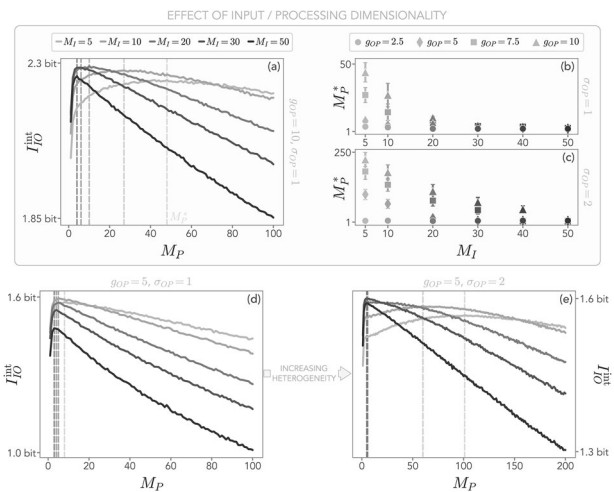

**Fig. 5 | Interplay between input and processing dimensionality. a** Mutual information between input and output $I_{IO}$ for nonlinear integration and a fast processing unit. For a given input dimension $M_I$, there exists an optimal processing dimensionality $M_P^*$ that maximizes the $I_{IO}$ (dashed lines). The coupling strength between processing and output is $g_{OP} = 10$. **b, c** Optimal processing dimensionality as a function of the input dimension $M_I$ for two different values of interaction heterogeneity $\sigma_{OP}$. Error bars represent one standard deviation over realizations of the random interaction matrices. The emergence of an optimal processing dimension is more evident at sufficiently strong couplings and is characterized by a decrease of $M_P^*$ as $M_I$ increases. The optimal processing dimension $M_P^*$ also increases with $\sigma_{OP}$. **d, e** Same as (**a**) for smaller $g_{OP}$ and different interaction heterogeneity $\sigma_{OP}$. At intermediate values ($g_{OP} = 5$) and low heterogeneity ($\sigma_{OP} = 1$), $I_{IO}^{int}$ is higher for smaller input dimensions and $M_P^*$ stays almost unchanged independently of $M_I$. Increasing $\sigma_{OP}$ to 2, large processing units become favored at small input dimensions and vice-versa. In this Figure, the standard deviations of the interaction matrices are $\sigma_{PI} = 1$, $\sigma_I = \sigma_P = 0.9$, and the input-processing coupling is $g_{PI} = 10$. Results are obtained by averaging over $2 \times 10^4$ realizations of the random interaction matrices.

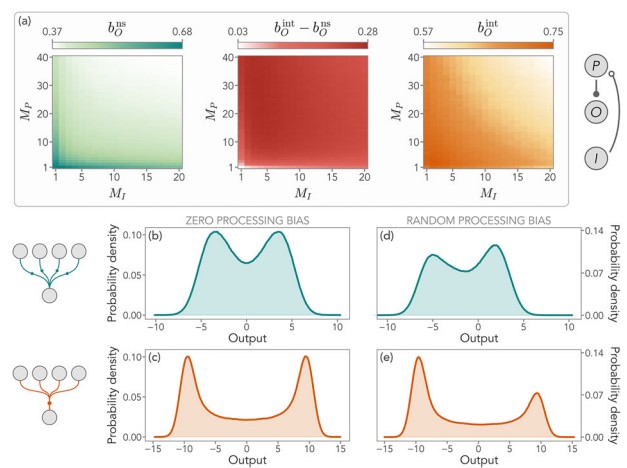

**Fig. 6 | Emergence of tunable bimodality in the presence of a fast processing unit. a** Bimodality coefficient of the output probability distribution, $b_O$, as a function of input and processing dimensions, respectively $M_I$ and $M_P$, in a system with a fast processing unit, as represented by the sketch next to the panel. We indicate the case of nonlinear summation in teal and with the superscript "ns", and the one of nonlinear integration in orange with the superscript "int". Here, $g_{PI} = g_{OP} = 10$, $\sigma_{OP} = \sigma_{PI} = 1$, and $\sigma_I = \sigma_P = 0.9$. Results are averaged over $10^3$ realization of random matrices. Both activation functions, implementing nonlinear summation or nonlinear integration, may lead to a bimodal output distribution, particularly for smaller dimensions. Notwithstanding, on average nonlinear integration enhances bimodality in all parameter ranges explored. **b, c** One-dimensional output distributions for $M_I = 5$ and $M_P = 10$ in the case of nonlinear summation (teal) and nonlinear integration (orange). **d, e** Same random matrix realization as in (**b, c**), but after introducing a random bias in the processing nonlinearity. The sketch next to (**b, c**) represents the two computational schemes considered. The bias allows for tuning the emergent output bimodality, allowing the system to select one of the modes. Importantly, due to its randomness, the bias may have opposite effects depending on the type of activation function.

implementing nonlinear summation, inputs are first nonlinearly transformed and then averaged, while inputs are first averaged and then transformed in systems supporting nonlinear integration. We showed that fast processing units outperform slow-processing ones, leading to higher input-output mutual information. Furthermore, we found that coupling strengths, interaction heterogeneity, and processing dimensionality— modulated by the number of connections between the units—determine whether nonlinear integration or summation is beneficial to the information shared between input and output units. Finally, we highlighted a nontrivial tradeoff between input and processing dimensions emerging in strong-coupling regimes.

Overall, our paradigmatic approach allowed us to investigate the emergence of accurate encoding in information-processing architectures. In particular, we highlighted the advantages of nonlinear integration in large multiscale systems and nonlinear summation in smaller ones. The information-theoretic differences between these two schemes may help in understanding optimal coding strategies and why certain biological systems implement integration or summation. The fact that nonlinear summation performs better with fewer degrees of freedom and more heterogeneous couplings may be especially relevant for those biochemical systems where a limited number of chemical species support signal propagation. Conversely, large neuronal networks may perform better by integrating incoming signals, with the specific topology of the interaction network determining the paths along which signals are dynamically propagated[8,9,69]. In this direction, a deeper understanding of the interplay between the nonlinearities and specific topologies is needed and has not been explored in this work, which has solely focused on the case of random interactions. A fundamental extension would be to use real-world networks as backbone structures to build processing layers. On the one hand, these investigations might

shed light on the ability of biological systems to process information; on the other hand, they might be informative for designing bio-inspired networks with optimal processing abilities. However, the design of information-processing systems faces the challenge of optimizing both nonlinear functions and network structure to implement given target functions. This goal will necessarily require the development of specific training algorithms that work for large stochastic systems. Similarly, studying scenarios with several output nodes is currently out of reach, since estimating the output entropy and mutual information in high dimensions is not feasible with the available numerical estimators.

Future works will need to consider other types of activation functions, evaluating their performances in terms of input-output information and the relative timescales between the units. Furthermore, it will be important to consider systems where different units implement different activation functions, allowing for more heterogeneity in terms of computational capabilities. In these scenarios, it will be interesting to systematically investigate how input, processing, and output dimensions shape information in specific real-world systems. Along this line, architectures with several processing units, possibly acting on a diverse range of timescales, may be necessary to deal with bio-inspired models and more structured inputs. This setting will also enable a natural implementation of multiple tasks whose presence might change the definition of processing performance, framing it in the context of decision-making and possibly connecting it with recent advances at the interface of information processing, decisions, and large language models across different scales[70]. Our work will stand as a fundamental step for these explorations, unraveling how different types of dynamical nonlinearities underlie information and computation in real-world systems.

## Methods

### Exact solution for fast processing units

For a system with a fast processing unit, the timescale separation $\tau_I \gg \tau_O \gg \tau_P$ leads to the steady-state or stationary joint probability distribution (i.e., the solution obeying $\partial_t p_{IPO}^{st} = 0$)

$$p_{IPO}^{fp} = p_{P|I}^{st} p_{O|I}^{eff,st} p_I^{st} \qquad (17)$$

where we omitted the superscript "st" on the l.h.s. for brevity, and

$$\begin{aligned} p_I^{st} &= \mathcal{N}_I(\mathbf{0}, \hat{\Sigma}_I) \\ p_{P|I}^{st} &= \mathcal{N}_P(\mathbf{m}_{P|I}(\mathbf{x}_I), \hat{\Sigma}_P) \\ p_{O|I}^{eff,st} &= \mathcal{N}_O(\mathbf{m}_{O|I}(\mathbf{x}_I), \hat{\Sigma}_O) \end{aligned} \qquad (18)$$

as we show explicitly in the Supplementary Note 4. The covariance matrices obey their respective Lyapunov equations, e.g., $\hat{A}_I \hat{\Sigma}_I + \hat{\Sigma}_I \hat{A}_I^T = 2\hat{D}_I$, with $\hat{D}_I = \text{diag}\left(D_I^1, \ldots, D_I^{M_I}\right)$, and similarly for the processing and output unit. Importantly, provided that the deterministic system is stable—i.e., that the eigenvalues of $\hat{A}_\mu$ have all positive real parts—these solutions exist and are well-defined[63]. Since we take $A_\mu^{ii} = 1$, when interactions are randomly distributed as $\mathcal{N}(0, \sigma_\mu / \sqrt{M_\mu})$, in the large $M_\mu$ limit, the $\mu$-th unit is stable if $\sigma_\mu < 1$[67]. The input-dependent mean of the processing, $\mathbf{m}_{P|I}(\mathbf{x}_I)$ is defined as in Eq. (8). Instead, the output distribution obeys an effective operator whose shape depends on whether nonlinear integration or nonlinear summation is employed. We find that, for nonlinear summation, the effective mean is given by

$$m_{O|I}^{ns,i}(\mathbf{x}_I) = \frac{g_{OP}}{M_P} \sum_{j=1}^{M_O} \sum_{k=1}^{M_P} \left(A_O^{-1}\right)^{ij} A_{OP}^{jk} \mathcal{F}\left(m_{P|I}^k(\mathbf{x}_I), \Sigma_P^{kk}\right) \qquad (19)$$

with

$$\begin{aligned} \mathcal{F}(x, v) &= \text{erf}\left(\frac{x}{\sqrt{2v}}\right) + \\ &+ \sum_{n=1}^{\infty} (-1)^n e^{2n^2 v} \left[V_n^+(x, v) - V_n^-(x, v)\right] \end{aligned} \qquad (20)$$

where we introduced the functions

$$\begin{aligned} V_n^+(x, v) &= e^{-2nx} \text{erfc}\left(\frac{2nv - x}{\sqrt{2v}}\right) \\ V_n^-(x, v) &= e^{2nx} \text{erfc}\left(\frac{2nv + x}{\sqrt{2v}}\right). \end{aligned} \qquad (21)$$

For nonlinear integration, the calculations are more involved. As reported in the main text,

$$m_{O|I}^{int,i}(\mathbf{x}_I) = g_{OP} \sum_{j=1}^{M_O} \left(A_O^{-1}\right)^{ij} \mathcal{F}\left(m_{int}^j \mathbf{x}_I, v_{int}^j\right) \qquad (22)$$

where

$$m_{int}^i(\mathbf{x}_I) = \frac{1}{M_P} \sum_{j=1}^{M_P} A_{OP}^{ij} m_{P|I}^j(\mathbf{x}_I). \qquad (23)$$

and

$$v_{int}^i = \frac{1}{M_P^2} \sum_{j=1}^{M_P} \sum_{k=1}^{M_P} A_{OP}^{ij} A_{OP}^{ik} \Sigma_P^{jk}. \qquad (24)$$

We present the detailed derivation of all these expressions in Supplementary Note 4.

### Exact solution for slow processing units

For a system with a slow processing unit, the timescale separation $\tau_I \gg \tau_P \gg \tau_O$ leads to the steady-state joint probability distribution

$$p_{IPO}^{sp} = p_{O|P}^{st}(\mathbf{m}_{O|P}(\mathbf{x}_P)) p_{P|I}^{st}(\mathbf{m}_{P|I}(\mathbf{x}_I)) p_I^{st} \qquad (25)$$

again omitting the superscript "st" on the l.h.s, with

$$\begin{aligned} p_I^{st} &= \mathcal{N}_I(\mathbf{0}, \hat{\Sigma}_I) \\ p_{P|I}^{st} &= \mathcal{N}_P(\mathbf{m}_{P|I}(\mathbf{x}_I), \hat{\Sigma}_P) \\ p_{O|P}^{st} &= \mathcal{N}_O(\mathbf{m}_{O|P}(\mathbf{x}_P), \hat{\Sigma}_O) \end{aligned} \qquad (26)$$

as we show explicitly in the Supplementary Note 4. With respect to the previous case, the crucial difference is that now $p_{O|P}^{st}$ is conditioned on the processing rather than on the input, due to the timescale structure. Furthermore, the input-dependent mean of the processing, $\mathbf{m}_{P|I}(\mathbf{x}_I)$ (see Eq. (8)) has the same form as the processing-dependent mean of the output, $\mathbf{m}_{O|P}(\mathbf{x}_P)$. We present the detailed derivation of all these expressions in the Supplementary Note 4.

### Direct input–output connections

We consider here the case of a system with only an input unit, $\mathbf{x}_I$, and an output unit, $\mathbf{x}_O$, with $M_O$ nodes. In the presence of a slow input, $\tau_I \gg \tau_O$, the steady-state or stationary solution of the Fokker-Planck equation for the joint probability distribution, $\partial_t p_{IO}^{st} = 0$, reads

$$p_{IO}^{np}(\mathbf{x}_I, \mathbf{x}_O) = p_I^{st}(\mathbf{x}_I) p_{O|I}^{st}(\mathbf{x}_O|\mathbf{x}_I) \qquad (27)$$

omitting "st" on the l.h.s., where

$$p_I^{st} = \mathcal{N}_I(\mathbf{0}, \hat{\Sigma}_I), \quad p_{O|I}^{st} = \mathcal{N}_O(\mathbf{m}_{O|I}(\mathbf{x}_I), \hat{\Sigma}_O) \qquad (28)$$

and

$$m_{O|I}^i(\mathbf{x}_I) = g_{OI} \sum_{k=1}^{M_O} \left(A_O^{-1}\right)^{ik} \phi_{OI}\left(A_{OI}^{k,1}, \ldots, A_{OI}^{k,M_I}; \mathbf{x}_I\right) \qquad (29)$$

for all $i = 1, \ldots, M_O$, as we explicitly show in the Supplementary Note 3.

### Exact sampling scheme for fast processing

In general, although our approach allows us to factorize the joint distribution $p_{IPO}^{st}$ into a product of Gaussian distributions for all timescale orderings, the input-output distribution, i.e., the one obtained from the marginalization over the processing dofs, remains highly nonlinear due to the nontrivial dependencies in the means of each Gaussian factor. However, our factorization allows for its efficient sampling, as all the nonlinearities appear as conditional dependencies. For a fast processing unit, for instance, we have

$$p_{IO}^{fp} = p_{O|I}^{eff,st} p_I^{st}$$

so that

$$h_{O|I} = \frac{1}{2}\left[M_O\left(1 + \log_2(2\pi)\right) + \log_2 \det \hat{\Sigma}_O\right] \qquad (30)$$

is easy to compute. Thus, we only need to evaluate $H_O$ numerically to estimate $I_{IO} = H_O - h_{O|I}$. The procedure for $N_{sam}$ samples is as follows:

1. sample $\{\mathbf{x}_I\}_{i=1}^{N_{sam}}$ from the independent Gaussian distribution of the input;
2. compute the means $\mathbf{m}_{O|I}(\{\mathbf{x}_I\}_i)$ for each sample $i$;

3. for all $i$, sample $\mathbf{x}_O$ from the multivariate Gaussian with covariance $\hat{\Sigma}_O$ and means $\mathbf{m}_{O|I}(\{\mathbf{x}_I\}_i)$.

Then, the entropy $H_O$ of the output distribution can be estimated from the samples $\{\mathbf{x}_O\}_i$ (see also Supplementary Note 4).

## Exact sampling scheme for slow processing

In the case of a slow processing unit, due to the timescale ordering, the conditional dependencies are crucially different. The input-output distribution,

$$p_{IO}^{\text{sp}}(\mathbf{x}_I, \mathbf{x}_O) = p_I^{\text{st}}(\mathbf{x}_I) \int d\mathbf{x}_P p_{O|P}^{\text{st}}(\mathbf{x}_O|\mathbf{x}_P) p_{P|I}^{\text{st}}(\mathbf{x}_P|\mathbf{x}_I), \quad (31)$$

cannot be easily computed. Thus, the entropy of the conditional distribution $h_{O|I}(\mathbf{x}_I)$ is not known analytically. To address this issue, we exploit the fact that we can efficiently sample $p_{O|I}^{\text{st}}$, allowing us to estimate directly the conditional entropy $H_{O|I} = \langle h_{O|I} \rangle_I$ with importance sampling. We detail below the sampling steps for a fixed number of input samples $N_{\text{sam},I}$ and $N_{\text{sam}}$ output samples per input:

1. sample a fixed input $\mathbf{x}_I^{(i)} \sim \mathcal{N}_I(0, \hat{\Sigma}_I)$ for $i = 1, \dots, N_{\text{sam},I}$;
2. for each input sample $\mathbf{x}_I^{(i)}$, compute $\mathbf{m}_{P|I}\left(\mathbf{x}_I^{(i)}\right)$, and extract the samples $\mathbf{x}_P^{(i,j)}$ from $\mathcal{N}_P\left(\mathbf{m}_{P|I}\left(\mathbf{x}_I^{(i)}\right), \hat{\Sigma}_P\right)$ for $j = 1, \dots, N_{\text{sam}}$;
3. for each processing sample $\mathbf{x}_P^{(i,j)}$, compute the mean $\mathbf{m}_{O|P}\left(\mathbf{x}_P^{(i,j)}\right)$ and extract the corresponding output $\mathbf{x}_O^{(i,j)}$ from $\mathcal{N}_O\left(\mathbf{m}_{O|P}\left(\mathbf{x}_P^{(i,j)}\right), \hat{\Sigma}_O\right)$;
4. for each input sample $\mathbf{x}_I^{(i)}$, estimate the entropy $h_{O|I}\left(\mathbf{x}_I^{(i)}\right)$ of the conditional distribution $p_{O|I}^{\text{st}}$ from the output samples $\{\mathbf{x}_O\}_{i,j}$;
5. estimate the conditional entropy $H_{O|I}$ via importance sampling.

Then, as in the case of fast processing, we can estimate the entropy $H_O$ from the output samples, and then the mutual information $I_{IO} = H_O - H_{O|I}$ (see also the Supplementary Note 2).

## Measures of bimodality

We measure the bimodality of the output distribution $p_O^{\text{st}}$ by computing Sarle's bimodality coefficient, defined as:

$$b = \frac{s^2 + 1}{\kappa + q(n_{\text{samples}})} \quad (32)$$

where $n_{\text{samples}}$ is the number of samples at hand, $q(x) = 3(n-1)^2/[(n-2)(n-3)]$, $s$ is the sample skewness,

$$s = \frac{1}{n_{\text{samples}}} \frac{\sum_i \left[x_O^{(i)} - \langle x_O \rangle\right]^3}{\left[\langle x_O^2 \rangle - \langle x_O \rangle^2\right]^{3/2}} \quad (33)$$

and $\kappa$ is the excess kurtosis,

$$\kappa = \frac{1}{n_{\text{samples}}} \frac{\sum_i \left[x_O^{(i)} - \langle x_O \rangle\right]^4}{\left[\langle x_O^2 \rangle - \langle x_O \rangle^2\right]^2} - 3. \quad (34)$$

## Code availability

The code employed to perform simulations and compute the input-output mutual information is available on Zenodo (https://doi.org/10.5281/zenodo.15324149).

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

## Acknowledgements

G.N. acknowledges funding provided by the Swiss National Science Foundation through its Grant CRSII5_186422. The authors acknowledge the

support of the Munich Institute for Astro-, Particle and BioPhysics (MIAPbP), funded by the Deutsche Forschungsgemeinschaft (DFG, German Research Foundation) under Germany's Excellence Strategy - EXC-2094 - 390783311, where this work was first conceived during the MOLINFO workshop. D.M.B. is funded by the program STARS@UNIPD with the project "ActiveInfo."

## Author contributions

G.N. and D.M.B. designed the study, performed calculations and numerical simulations, interpreted the results, and wrote the manuscript.

## Funding

## Competing interests

The authors declare no competing interests.
