## [Transparent Peer Review file · Communications Physics]

Fast nonlinear integration drives accurate encoding of input information in large multiscale systems

Corresponding Author: Dr Daniel Maria Busiello

This manuscript has been previously reviewed at another journal. This document only contains information relating to versions considered at Communications Physics.

Version 0:

Reviewer comments:

Reviewer #1

(Remarks to the Author)

In their manuscript entitled "Fast nonlinear integration drives accurate encoding of input information in large multiscale systems", the authors take an information theoretic approach at information processing in large systems. This manuscript has already undergone rounds of review, so the main point of this assessment are to identify outstanding issues that need to be addressed by the authors.

(i) Referees 3+4 raised questions about the generality of the results. I agree with the referees that some clarification is lacking. The authors write "to obtain general results" but then limit their attention to specific random networks where connectivity is drawn from certain distributions. This raises the question whether specific networks (e.g., with a prescribed degree distribution) have similar properties. From the best of my understanding, this was the main point of the referees—rather than incorporating any additional dynamics, "training", or similar. So to me a key question is indeed to understand how the assumptions at the bottom of p4 affect the results.

(ii) I believe the the authors have mostly addressed the comments by Referee 5. Of course, the authors should be mindful of overly many self-citations, especially in the context of a number of papers/preprints on related topics by the same authors (plus epsilon) in a short time period.

(iii) The authors claim that they have toned down "grandiose language". I think the authors should have another serious look at their manuscript given that they write "Our results uncover the pivotal features of nonlinear information processing with implications for both biological and artificial systems." already in their abstract, which is quite a statement. That goes hand in hand with point (i), so it would be good if the authors revisited their language, claims of generality, and discuss limitations.

Version 1:

Reviewer comments:

Reviewer #1

(Remarks to the Author)

In their revision, the authors made additions to understand their results in the context of different network topologies. While this goes in the right direction, it does have a feeling of "minimal changes required" - also because the discussion stays vague. Sparsity, according to the figure caption, means that it is still uniform (not general sparse networks) and Barabasi-Albert is just one potential nontrivial network structure (why this example is not clear to me).

An honest assessment and (somewhat more extended) discussion should be added to the manuscript. This would be helpful to put results in context and point out limitations. In particular, the added results by the authors - unless they give convincing arguments otherwise - should be portrayed in a factual way for what they appear to be: A limited exploration into

different network structures. The authors should be critical and avoid overselling.

REVIEWER #1

In their manuscript entitled "Fast nonlinear integration drives accurate encoding of input information in large multiscale systems", the authors take an information theoretic approach at information processing in large systems. This manuscript has already undergone rounds of review, so the main point of this assessment are to identify outstanding issues that need to be addressed by the authors.

We thank the reviewer for the time dedicated to review our manuscript, which considerably helped improving our manuscript. Please find below a point-by-point answer to all their concerns.

(i) Referees 3+4 raised questions about the generality of the results. I agree with the referees that some clarification is lacking. The authors write "to obtain general results" but then limit their attention to specific random networks where connectivity is drawn from certain distributions. This raises the question whether specific networks (e.g., with a prescribed degree distribution) have similar properties. From the best of my understanding, this was the main point of the referees—rather than incorporating any additional dynamics, "training", or similar. So to me a key question is indeed to understand how the assumptions at the bottom of p4 affect the results.

Our impression was that the previous referees #3-#4 were explicitly referring to training. However, since this is not feasible in large stochastic systems such as our own, we believe that the referee's suggestion to use different topological structures is a very good idea to check the robustness of our result in this direction without invoking training or optimality. We thank them for this insight.

In the revised version, Figure 4 has been extended. We now study the effect of sparsity of connections between the units, finding - as expected - that more sparse units are qualitatively similar to units in lower dimension. This is because the number of incoming connections from a unit to another effectively decreases. Importantly, the trend with the processing dimension highlighted in Figure 4 remains valid, i.e., larger dimensions favor integrations across coupling regimes.

Furthermore, as suggested by the referee, we have studied the case of a system where the intra-unit topology of the processing unit is described by a Barabasi-Albert network with random weights. We find that our results are qualitatively robust with respect to this change, and that this structure actually tends to improve the mutual information when integration is used (Figure 4f). We believe that these examples strengthen the fact that the properties we see do not depend on the specific network choice.

(ii) I believe the the authors have mostly addressed the comments by Referee 5. Of course, the authors should be mindful of overly many self-citations, especially in the context of a number of papers/preprints on related topics by the same authors (plus epsilon) in a short time period.

Following our response to Referee 5, we also believe it is important to include only references that are directly relevant to the content of the present study. Indeed, in the previous round of review, we reviewed all self-citations and removed some of marginal importance during the first round of revision. At this stage, we believe the remaining references are pertinent to different aspects of the work, and have therefore decided to retain them. If the reviewer feels that their number should be further reduced, we may need to eliminate several additional references, keeping only those strictly related to the submitted manuscript. However, we preferred to provide the reader with a broader perspective in this respect.

(iii) The authors claim that they have toned down "grandiose language". I think the authors should have another serious look at their manuscript given that they write "Our results uncover the pivotal features of nonlinear information processing with implications for both biological and artificial systems." already in their abstract, which is quite a statement. That goes hand in hand with point (i), so it would be good if the authors revisited their language, claims of generality, and discuss limitations.

We modified some sentences in the abstract and discussion, as they might have sounded excessive relative to the actual content of the manuscript. Our initial intention in including them was to convey a message and potential impact beyond the specific results of the study. However, we agree with the referee that such language might be counterproductive or unrealistic, and we have therefore decided to follow their suggestion.

REVIEWER #1

In their revision, the authors made additions to understand their results in the context of different network topologies. While this goes in the right direction, it does have a feeling of "minimal changes required" - also because the discussion stays vague. Sparsity, according to the figure caption, means that it is still uniform (not general sparse networks) and Barabasi-Albert is just one potential nontrivial network structure (why this example is not clear to me). An honest assessment and (somewhat more extended) discussion should be added to the manuscript. This would be helpful to put results in context and point out limitations. In particular, the added results by the authors - unless they give convincing arguments otherwise - should be portrayed in a factual way for what they appear to be: A limited exploration into different network structures. The authors should be critical and avoid overselling.

We thank the reviewer for taking the time to review our manuscript again. We would like to point out that the focus of our addition was to validate the robustness of our results (at least at a qualitative level) for a different network topology. The original aim of our manuscript was primarily to study the information-theoretic differences associated with different nonlinear operations for a fixed network structure and random interactions. In particular, to eliminate the variability given by the topology, we originally used fully connected networks for the processing layer.

Nevertheless, we have now added other topologies (Erdos-Renyi and small-world networks), for different parameters, in the Supplementary Information. The results remain qualitatively unchanged, suggesting once more that the sparsity of the processing unit favors nonlinear integration in terms of mutual information. We also showed that different choices for the connection probabilities between the nodes of two different units do not affect our results, as we argued in the previous round (sparsity between the units is akin to reducing the dimensionality). We also recall that the units have different dimensions, i.e., different numbers of nodes, so the connections between them are described by a rectangular matrix and not a square one. If the referee has in mind any particular network structure or probability of connection between the units that they believe will drastically change our results, we would be happy to test it.

We believe that our results strongly point to the fact that information mostly depends on the form of nonlinearities, not on the specific network structure, at least in the presence of random couplings. Of course, we agree on the fact that topology might play a crucial role in real-world scenarios with specific interactions, possibly optimized for some function. This kind of study exceeds the scope of our manuscript, whose intention was to give a first general analytical approach to nonlinear processing across timescales. To better emphasize all these aspects, we modified some sentences in the manuscript.

We hope that our changes in the revised version clarify the scope of our exploration and give a better intuition of their robustness. We believe that, without specific examples in mind, we may end up with a zoology of similar scenarios from which it will be very difficult to draw clear conclusions. Studies more focused on real-world examples will be the subject of future investigations.